# Effect of price on the decision to experiment with cigarette smoking among Gambian children: a survival analysis using the Gambia 2017 Global Youth Tobacco Survey data

Chengetai Dare ![ORCID],[1] Bai Cham ![ORCID],[2] Micheal Kofi Boachie ![ORCID],[3] Zachary Gitonga,[1] Umberto D'Alessandro,[2] Corne Walbeek[1]

[1]Research Unit on the Economics of Excisable Products (REEP), School of Economics, University of Cape Town, Rondebosch, Western Cape, South Africa
[2]Medical Research Council Unit The Gambia School of Hygiene and Tropical Medicine, Banjul, Gambia
[3]SAMRC/Wits Centre for Health Economics and Decision Science -PRICELESS SA, University of the Witwatersrand Faculty of Health Sciences, Johannesburg, South Africa

**Correspondence to**
Dr Chengetai Dare;
cdarejam@yahoo.com

## ABSTRACT

**Objectives** To investigate the relationship between cigarette prices and smoking experimentation among children in the Gambia, and thereby expanding the evidence base of the likely impact of excise taxes on cigarette demand in low-income and middle-income countries.

**Design** A survival analysis using the Gambia 2017 Global Youth Tobacco Survey data.

**Setting** The Gambia.

**Participants** The survey sample was 12 585 youths, aged 12–17 years, but our analysis was restricted to 11 030 respondents with information on smoking status.

**Primary and secondary outcome measures** Our outcome variable was smoking experimentation defined as the first time the respondent smoked (at least part of) a cigarette.

**Results** A 1% increase in the price of cigarettes reduces the probability to experiment with smoking by 0.7%. We also found that children are more likely to experiment with smoking if they have at least one smoking parent, friends who smoke and see teachers who smoke. The probability to experiment with cigarette smoking increases with age and is higher among boys than girls.

**Conclusion** There is strong evidence that increasing excise taxes can play an effective role in discouraging children from experimenting with cigarette smoking. Considering the relatively low excise tax burden in the Gambia, the government should consider substantially increasing the excise tax burden.

## STRENGTHS AND LIMITATIONS OF THIS STUDY

⇒ Survival analysis using pseudolongitudinal data (obtained from cross-sectional data) is a good strategy in estimating longitudinal outcomes in settings where longitudinal studies are costly and/or difficult to conduct.

⇒ There are very few tobacco demand studies in sub-Saharan Africa that employed survival analysis, and this is the first in the Gambia.

⇒ The Global Youth Tobacco Survey is cross-sectional and focuses on students in junior secondary schools, the data may not be a good representation of Gambian youth, including those in senior secondary schools, children out of school and school drop-outs.

⇒ The study relied on individuals' own memory regarding past smoking behaviour and hence the dependent variable could be subject to measurement error (specifically, recall bias).

## INTRODUCTION

Smoking is the leading cause of preventable death.[1] There are over one billion smokers globally and around 80% of them live in low-income and middle-income countries (LMICs).[1 2] Africa's share of the global number of smokers is rapidly increasing.[3] Over 8 million people die every year from tobacco use and/or exposure to tobacco smoke,[2 4] and 80% of all tobacco-related deaths by 2030 are predicted to occur in LMICs.[5 6] Tobacco companies are shifting their target to LMICs to build a broader consumer base, especially among young people.[7 8] In the WHO African region, the total number of smokers is projected to increase by 27.3% from 66 million in 2015 to 84 million by 2025[9]; this is the second-largest percentage increase after the Eastern Mediterranean region.

Research has shown that experimentation with cigarettes among young people is associated with future daily smoking.[10 11] In a study among 6522 US adolescents aged 10–14 years, experimentation with cigarette smoking during ages 10–14 years predicted daily smoking 2 years later, independent of other cigarette smoking risk factors.[10] Evidence has also shown that higher prices is a deterrent of cigarette use among young people, especially in Africa. In several African countries,

higher cigarette prices have contributed to a reduction of both tobacco use prevalence and intensity.[12] This has been confirmed in country-specific studies in Sierra Leone,[13] South Africa,[14] Tanzania,[15] Kenya,[16] Zambia[17] and Uganda.[18] Similarly, studies have found that, in Ghana and Nigeria, increased cigarette prices decreased both smoking intensity and smoking initiation.[19]

The Gambia is the smallest country on mainland Africa with a population of about 2 million. It is a low-income country and is ranked 172 out of 189 countries in the 2020 United Nations Human Development Index,[20] with per capita Gross Domestic Product (GDP) of US$773.[21] The country ratified the WHO Framework Convention on Tobacco Control in September 2007.[22] There are several regulations and policies on tobacco control in The Gambia, including the Prohibition of Smoking in Public Places Act 1998, the Tax Policy Reform Act 2013, the Tobacco Control Act 2016 and the Tobacco Control Regulations 2019.[23 24] These legal instruments seek to protect children from exposure to tobacco products. However, despite the numerous achievements in tobacco control as highlighted above, there are still gaps, especially on the implementation of smoke-free regulation in the Gambia.[24] There has also been some regression, specifically with the Tax Policy Reform Act, which has been stalled since 2017, following a change in government.

The prevalence of smoking is 15.9% among Gambian adults (25–64 years),[25] 1% among women and 32.1% among men.[26] A 2016 survey conducted among secondary school students (12–20 years) indicated that 7.9% of boys and 1.5% of girls had smoked at least once in the past 30 days.[27] An earlier 2008 Global Youth Tobacco Survey (GYTS) survey indicated that 10.8% of 13–15 years students (12.7% among boys and 8.6% among girls) had smoked at least once in the past 30 days.[28]

Like other governments across the world, the Gambian government employs tax and price measures to reduce the affordability of tobacco. For instance, in 2013, a specific excise tax was introduced on all imported tobacco products. This contributed to an increase in cigarette prices and tax revenue and to a reduction in tobacco imports.[29] However, cigarette prices remain relatively low. Using 2019 exchange rates, data obtained from The Gambia Bureau of Statistics[30] show that the average retail price for a packet of 20 sticks (of the most popular brands) ranged between GMD26.15 (US$0.51) and GMD31.70 between 2008 and 2019, which is substantially lower than the sub-Saharan African regional average of US$1.80, and the global average price of US$3.82.[31] The tax burden (share of taxes in the retail price) was 46.3% in 2018,[31] which is below the 75% target recommended by the WHO.[32]

An increase in the excise tax is the single most important intervention to reduce smoking prevalence and intensity.[32 33] Youth are even more price-sensitive than adults, making tobacco taxation particularly effective; this is even more so in countries with a young population.[19] Although global knowledge about the effect of prices (or taxes) as a tobacco control measure is well established, according

to our knowledge, there is no evidence on the impact of prices on tobacco use experimentation among the children in the Gambia. In this paper, we fill this knowledge gap by employing survival analysis to examine the effect of prices on the decision to experiment with cigarette smoking among children in the Gambia.

## Data and methodology

We used the 2017 GYTS[34] to obtain individual and household information on smoking behaviour and other background characteristics. The GYTS is a nationally representative school-based survey designed by the Centers for Disease Control and Prevention (CDC) as a global standard tool for monitoring tobacco use among youth and to guide the implementation and evaluation of tobacco prevention and control programmes.[34] The survey considers a cross-section of students in junior secondary schools (grades 7–9). It does not follow individuals over time, but provides a snapshot on their smoking patterns. The sample is drawn using a two-stage cluster-sampling design.[35 36] Schools are selected with probability proportional to school enrolment size during the first stage, and then classes within participating schools are selected as a systematic equal probability sample with a random start during the second stage.[36] All students in the selected classes are eligible to participate in the survey.[34 36]

The 2017 GYTS covers a sample of 12 585 youths, aged 12–17 years. However, 1555 respondents did not provide sufficient information about their smoking status, thereby reducing our sample size to 11 030. Smoking experimenters or ever-smokers (defined as having smoked at least once or twice over their lifetimes) were reported by 2218 (20.1%) respondents (table 1).

The study employs a duration (or survival) model to estimate the probability of a respondent experimenting with smoking. Survival analysis allows us to analyse the length of time until the occurrence of a well-defined end point of interest,[37] which in this case is smoking experimentation. Experimentation is defined in terms of the first time the respondent smoked (at least part of) a cigarette. It is obtained from the question: 'Have you ever tried or experimented with cigarette smoking, even one or two puffs?'. The timing of the transition from having never smoked into experimentation depends on the probability of experiencing a transition in period $t$, conditional on not having experienced a transition until period $t$; is also known as the hazard rate or conditional failure rate.[16] The estimation approach requires longitudinal data, making it necessary to transform the cross-sectional GYTS data into pseudolongitudinal data.[16 19 38] The transformation allows for the analysis of time-to-event data. Such data describe the length of time until the occurrence of the event of interest.

As in Asare et al,[19] to obtain the longitudinal data required for the duration analysis, we retroactively inferred the year of smoking experimentation using the GYTS question on the age of smoking experimentation: 'How old were you when you first tried a cigarette?'. For

**Table 1** Demographic characteristics and smoking habits (%)

| | N (and proportion of total sample) | Ever-smokers in the category |
|---|---|---|
| Students sampled (N=11 030) | n=11 030 | n=2218 (20.1%) |
| Gender (n=10 886) | | |
| Male | 44.3 | 32.9 |
| Female | 55.7 | 9.6 |
| Age (n=10 483) | | |
| 12 | 6.0 | 16.4 |
| 13 | 14.9 | 15.0 |
| 14 | 23.6 | 16.7 |
| 15 | 21.7 | 18.3 |
| 16 | 17.1 | 22.4 |
| 17 | 16.7 | 28.1 |
| Parent(s) smoke(s) (n=10 945) | 18.4 | 29.7 |
| Friend(s) smoke(s) (n=10 916) | 23.0 | 38.0 |
| Amount of pocket money per week (n=10 918): | | |
| Usually do not have | 12.1 | 17.4 |
| Less than GMD25 | 35.7 | 19.0 |
| GMD25–GMD50 | 32.5 | 19.8 |
| GMD51–GMD100 | 10.5 | 23.1 |
| GMD101–GMD150 | 3.9 | 23.0 |
| GMD151–GMD200 | 1.9 | 25.0 |
| Above GMD200 | 3.4 | 27.8 |
| Sees teachers smoking in school buildings | 29.7 | 25.8 |
| Regards quitting smoking as difficult (n=10 937) | 37.4 | 21.8 |
| Mean age (years) | 14.8 (SD 1.47) | 15.1 (SD 1.50) |
| Mean experimentation age (years) | | 11.4 (SD 2.96) |

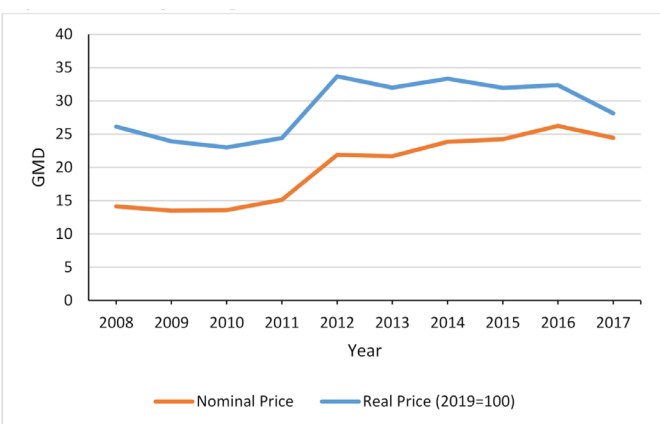

**Figure 1** Average real price, 2008–2017.

For regression analysis, our baseline model uses a logistic distribution for the hazard function because of its flexibility to allow non-monotone, time-dependent changes of the hazard rate.[16 42] The regression model estimates the hazard of initiation for individual $i$ in period $t$ ($y_{it}$) as a function of real cigarette prices ($price_t$) and a matrix of explanatory variables ($X_i$), which include sociodemographics (age, gender, parents' and friends' smoking status, income (pocket money) and respondents' perception on quitting smoking). Gender takes the value of 1 for female and 0 for male. Smoking status of the respondent's parents and friends takes the value of 1 if at least one of the parents or friends smokes, and 0 if none of the parents or friends smoke. The respondents' income (pocket money) is derived from the question: 'During an average week, how much money do you have that you can spend on yourself, however you want?'. The base is 'I usually don't have', and the six other categories are 'Less than GMD25', 'GMD25-GMD50', 'GMD51-GMD100', 'GMD101-GMD150', 'GMD151-GMD200' and 'Above GMD200'. Respondents' perception on quitting smoking is derived from the GYTS question: 'Once someone has started smoking tobacco, do you think it would be difficult for them to quit?' The variable is coded as 1 for 'yes' and 0 for 'no'. The dependent variable, that is, cigarette smoking experimentation, is extracted from the GYTS data.[34]

The regression model is specified as:

$$Y_{it} = Pr(\textit{initiate}|\textit{no prior smoking})$$
$$= \beta_1 price_t + \beta_2 X_{it} + \varepsilon_{it} \qquad (1)$$

We tested the robustness of the results by incorporating a discrete-time split population survival model, which relaxes the assumption that all individuals will eventually smoke. It first estimates each individual's probability of ever experiencing a smoking transition, then weights the hazard function by this probability. The contribution of individual $i$ to the log-likelihood function of smoking experimentation is:

each age, the individuals were assigned a value of 0 if they did not experiment. The person is assigned a value of 1 in the year in which they tried a cigarette and then drops out of the dataset. The pseudolongitudinal dataset was constructed on the assumption that a person is at risk of having his/her first cigarette at the age of 8.[19 38–41] The eldest among the respondents was 17 when the survey was conducted in 2017. These respondents were 8 years in 2008. As such, our data sample excluded those who started experimenting with cigarettes before 2008. We also excluded respondents that reported ever-smoking, but who did not provide sufficient data on when they experimented with smoking.

The pseudolongitudinal data are then merged with the national-level cigarette price, which allows us to investigate the relationship between price and smoking experimentation. The price data were obtained from The Gambia Bureau of Statistics. The price trend is shown in figure 1.

**Table 2** Regression results: logistic and split population survival models

| | Logistic regression model | | Split population model |
|---|---|---|---|
| | Coefficients | ORs | HRs |
| | (1) | (2) | (3) |
| Price elasticity of experimentation | −0.680** | 0.507** | 0.524*** |
| | (0.302) | (0.153) | (0.131) |
| Parent(s) smoke(s) | 0.371*** | 1.449*** | 1.474*** |
| | (0.082) | (0.120) | (0.093) |
| Friend(s) smoke(s) | 0.482*** | 1.620*** | 1.828*** |
| | (0.073) | (0.119) | (0.102) |
| Difficulty of quitting | 0.055 | 1.056 | 1.114* |
| | (0.069) | (0.073) | (0.062) |
| Teachers smoke in school buildings | 0.176** | 1.192** | 1.205*** |
| | (0.073) | (0.087) | (0.068) |
| Usually do not have pocket money | 0.000 | 1.000 | 1.000 |
| Less than GMD25 | 0.083 | 1.087 | 1.113 |
| | (0.127) | (0.138) | (0.109) |
| GMD25–GMD50 | 0.067 | 1.070 | 1.181* |
| | (0.125) | (0.134) | (0.114) |
| GMD51–GMD100 | 0.334** | 1.397** | 1.400*** |
| | (0.141) | (0.197) | (0.158) |
| GMD101–GMD150 | 0.080 | 1.084 | 1.224 |
| | (0.190) | (0.206) | (0.190) |
| GMD151–GMD200 | 0.246 | 1.279 | 1.396* |
| | (0.227) | (0.290) | (0.261) |
| Above GMD200 | 0.411** | 1.508** | 1.505*** |
| | (0.181) | (0.273) | (0.224) |
| Female | −1.403*** | 0.246*** | 1.159*** |
| | (0.083) | (0.020) | (0.016) |
| Age | 0.165*** | 1.179*** | 1.159*** |
| | (0.020) | (0.024) | (0.016) |
| Constant | −3.426*** | 0.033*** | 0.028*** |
| | (0.892) | (0.029) | (0.021) |
| Observations | 77 640 | 77 640 | 77 640 |

SE in parentheses.
*p<0.10; **p<0.05; ***p<0.01.

$$d_i = ln\left\{P_r\left(ever\ initiate\right) * f\left(t|t>0\right)\right\} + \left(1 - d_i\right)$$
$$* ln\left\{P_r\left(ever\ initiate\right) + P_r\left(ever\ initiate\right) * f\left(t|t=0\right)\right\} \quad (2)$$

where $d_i$ is a binary indicator for experimenting with smoking at some point during the period of observation, $t$ is the time of experimentation measured in number of years since age eight, and $f(t)$ is its probability density function.

As in Vellios and van Walbeek,[38] we used Stata's *spsurv* command for the split population estimation model. The model has been used in numerous studies.[38 43 44] The *spsurv* command uses a complementary log-log specification, which reports hazard ratios (not ORs) and unlike the logit specification, the response curve is asymmetric.[42] The ORs are calculated by taking the antilog of the coefficient. The hazard rate is calculated as (OR/(1+OR) where OR is the odds ratio. All analyses were done with Stata V.15.

## Patients and public involvement
Patients and the public were not involved in this study.

## RESULTS
The regression results are shown in table 2. The estimated ORs and HRs are in exponentiated form and are broadly similar. The HRs are indicated in the table but are not discussed in this section.

The results show that a 1% increase in the price of cigarettes reduces the probability to experiment with smoking among children by 0.7% (95% CI −1.1% to −0.4%) (as indicated in column 1).

Children with at least one smoking parent are more likely to have ever smoked a cigarette than those whose parents do not smoke (OR 1.4, 95% CI 1.2 to 1.7). Also, children with friends who smoke are more likely to have ever experimented with smoking a cigarette than those whose friends do not smoke (OR 1.6, 95% CI 0.4 to 1.9).

Children who see their teachers smoking inside school buildings are more likely to have ever experimented with smoking a cigarette than those who do not see teachers smoking inside school buildings (OR 1.2, 95% CI 1.0 to 1.4). Income (ie, having access to pocket money) did not have a clear effect, although there is some suggestion that children who receive pocket money are more likely to experiment with smoking than children who do not receive pocket money.[45]

Female youths are less likely to have ever smoked a cigarette than male youths (OR 0.2, 95% CI 0.2 to 0.3). The results also show that a 1-year increase in age increases the probability to experiment with smoking (OR 1.2, 95% CI 1.1 to 1.2). Figure 2 shows the smoking initiation hazard rates by age and gender. The hazard rates are derived from the regression analyses.

Figure 2 shows a positive correlation between age and the probability to experiment with smoking cigarettes among the youth in the Gambia. The probability to experiment with smoking is higher among boys than among girls.

## DISCUSSION
The price elasticity of demand for smoking experimentation is estimated to be −0.7 among children in The Gambia. This implies that a 10% rise in the price of cigarettes is associated with a 7% reduction in the probability

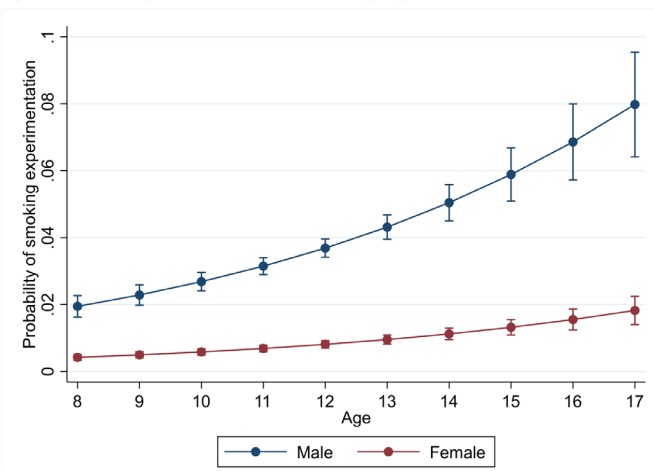

**Figure 2** Smoking hazard rates for smoking experimentation.

to experiment with smoking among the Gambian children. Increasing excise taxes could be a highly effective tobacco-control tool in The Gambia. The excise tax typically increases the retail price of cigarettes, thereby reducing the propensity to experiment with smoking.[1 16] Considering that the current tax burden of 46.3% is well below the 75% target recommended by the WHO,[46] there is room for the government to further raise the excise tax on tobacco. Raising tobacco tax is a 'win-win' policy, as it generates extra revenue for the government while discouraging tobacco consumption through higher prices, thereby improving public health.[29]

The study established that children with smoking friends were more likely to have ever smoked a cigarette than those whose friends do not smoke. Similarly, children from households with at least one parent who currently smokes were more likely to have ever smoked a cigarette. This could be because, as children are more exposed to smoking (especially by parents) they would regard smoking as normal. Also, if the parents smoke, access to cigarettes is easier, than if the parents do not smoke. The results from this study are in line with the findings of Jallow *et al*[47] who found that children in smoking families were more likely to smoke than those in non-smoking families. We also found that children are more likely to experiment with cigarette smoking if they see their teachers smoking in school buildings. If smoke-free regulations are effectively implemented, it would presumably reduce smoking in school buildings, and thus reducing youth smoking. Nevertheless, our study found that peer influence was more important in experimenting with smoking than both parental and teachers' influence. A systematic review of studies conducted among high school and university students in Ethiopia and other studies in the USA found that peer pressure had a significant influence on young peoples' initiation and smoking of cigarettes.[33 48] Increases in the price will have a direct and indirect influence on smoking. The direct influence will be through the price elasticity of demand. The

indirect influence will be through the peers. If the price increases, the peers will smoke less (or not at all), thus decreasing the peer pressure.

The Gambian youths start smoking as early as at age eight, a result similar to a previously published study.[26] The likelihood of smoking increases exponentially as age increases. As in most other countries, the probability of smoking is higher among boys than among girls. This is mirrored by the smoking prevalence in the general population, where smoking prevalence among women is very low (about 1%) compared with men (32.1%).[26] However, the smoking rate among girls is substantially higher than among adult women, suggesting that female smoking prevalence may increase sharply in future, as girls step into adulthood. This would be detrimental to public health. The government should implement policies, including an increase in the excise tax, that will prevent an increase in smoking experimentation.

## Limitation of the study

Although this study provides useful information for devising suitable tobacco-tax policy measures, there are some limitations to consider. For instance, the study relied on individuals' own memory regarding past smoking behaviour. As such, the dependent variable could be subject to measurement error with an unknown bias. However, this challenge is more prevalent among adult respondents.[16 19] Considering that our sample was composed of children, with relatively short smoking histories, we believe that our findings are robust. In addition, the study relied on a pseudopanel where several individual demographic and socioeconomic characteristics were fixed at the time of the survey. To the extent that some of these socioeconomic variables have changed in the years prior to the survey, this could have biased the results. Also, considering that the 2017 GYTS is a cross-sectional survey among students in junior secondary schools, the data may not be a good representation of Gambian youth, for example, those in senior secondary schools (grades 10–12) and school drop-outs. We do not have information on what happens beyond this age group, because the data are not collected. In addition, some important information such as the respondents' region, race and religion were not collected. Had these variables been available, they could have been used in the regression analysis.

## CONCLUSION

We found the price elasticity of demand for smoking experimentation to be −0.7 among children who have smoked at least once or twice in their lives. Our results indicate that an increase in tobacco excise taxes can play an effective role in discouraging children from experimenting with smoking. Considering the current relatively low excise tax burden, the government should consider increasing the excise tax burden on tobacco products in line with the recommendations of the WHO.[32]

**Acknowledgements** Special thanks to Nicole Vellios who reviewed earlier drafts.

**Contributors** CD and MKB conceptualised the study and conducted the data analysis. CD, MKB and BC wrote the first draft of the manuscript. CW, ZG and UD'A contributed to the drafting and revision of the manuscript. CvW is responsible for the overall content as guarantor.

**Funding** This study was primarily supported by the UK Research and Innovation (UKRI) with funding from the Global Challenges Research Fund (MR/P027946/2), as part of the Tobacco Control Capacity Programme (TCCP). The TCCP is a programme of capacity development and research co-ordinated by the University of Edinburgh, Scotland and involves 15 partner institutions from Africa, South Asia and the United Kingdom. The study was also supported by the Bill and Melinda Gates Foundation through the African Capacity Building Foundation (grant no: 334) and by the South African Medical Research Council (grant no: 23108) through the SAMRC/Wits Centre for Health Economics and Decision Science - PRICELESS SA.

**Competing interests** None declared.

**Patient and public involvement** Patients and/or the public were not involved in the design, or conduct, or reporting, or dissemination plans of this research.

**Patient consent for publication** Not applicable.

**Provenance and peer review** Not commissioned; externally peer reviewed.

**Data availability statement** Data are available in a public, open access repository. Data are publicly available on: https://nccd.cdc.gov/GTSSDataSurveyResources/Ancillary/DataReports.aspx?CAID=2.

**ORCID iDs**
Chengetai Dare http://orcid.org/0000-0001-5001-8768
Bai Cham http://orcid.org/0000-0002-1656-2126
Micheal Kofi Boachie http://orcid.org/0000-0003-1062-889X

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
