## [Reviewer comments · BMJ Open]

ARTICLE DETAILS

TITLE (PROVISIONAL)	The effect of price on the decision to experiment with cigarette smoking among Gambian children: A survival analysis using the Gambia 2017 Global Youth Tobacco Survey data
AUTHORS	Dare, Chengetai; Cham, Bai; Boachie, Micheal Kofi; Gitonga, Zachary; D'Alessandro, Umberto; Walbeek, Corne

VERSION 1 – REVIEW

REVIEWER	Ma, Shaoying OSUMC, Center for Tobacco Research
REVIEW RETURNED	04-May-2022

GENERAL COMMENTS	It is with pleasure that I review the paper titled “The effect of price on the decision to experiment with cigarette smoking among Gambian children”. The authors use the method of survival analysis, to study the link between cigarette prices and youth (12-17 yo) experimenting with smoking in The Gambia. Data used in this paper include Global Youth Tobacco Survey (GYTS) data, as well as data from the Gambia Bureau of Statistics. The authors find that 1% increase in cigarette price is associated with 0.7% decrease in the likelihood of experimenting with smoking (defined as taking first puff) among youth. They also document other important factors of youth experimenting with smoking in The Gambia, aside from cigarette prices. Furthermore, this study shows gender differential in the behavior of experimenting with smoking among Gambian youth. This study has the important policy implication that raising excise tax on cigarettes in The Gambia could be effective in preventing youth from experimenting with smoking. Although there have been studies on the prevalence and factors of cigarette smoking (or tobacco use in general) among adults and youth in The Gambia, this study contributes to the literature by examining cigarette price as a significant factor of Gambian youth experimenting with smoking, and it has important policy implication on increasing taxation to prevent youth initiation of cigarette use. Please see below for specific comments. The page numbers are based on those at the lower right corner of the manuscript. 1. Page 4 line 21 “... this is the second-largest percentage increase after the Eastern Mediterranean region.” Consider rephrasing this sentence to make it more readable, especially because the percentage change was not reported in the sentence.
--

	2. Page 5 line 3 “The Gambia is a low-income country and is ranked 172 out of 189 countries in the 2020 United Nations Human Development Index.” For the ranking in UN human development index, add citation. 3. Page 5 line 3 “The country ratified the WHO Framework Convention on Tobacco Control (WHO FCTC) in September 2007. There are several regulations and policies on tobacco control in The Gambia, including the Prohibition of Smoking in Public Places Act 1998, the Tobacco Control Regulations 2019 and the Tobacco Control Act 2016.” Citations need to be added to both sentences. 4. Page 6 line 30 How is junior secondary school defined in The Gambia? 5. Page 6 line 32 I think the authors could add some information here to talk about the representativeness of GYTS sample. Is it a nationally representative sample of junior secondary school students in The Gambia? What is its sampling strategy? Is it a school-based survey? 6. Page 6 Table 1 column 2 Is there a reason why 55.7% are girls and 44.3% are boys in the sample of youth with non-missing information on their smoking status? 7. Page 7 line 34 The rationale for choosing survival model as the method of this paper should be provided in the methodology section. Also consider providing that at the end of the introduction section. What is the advantage of using survival analysis, and why is it appropriate for the research question and the data you use? Please discuss those briefly. 8. Page 8 line 34 More details about the price data from the Gambia Bureau of Statistics need to be provided. Are those data publicly accessible? How was the cigarette price measure calculated by the Bureau of Statistics? Is it the price of the most popular brand, or something else? Is it the price of 20 sticks in one pack of cigarette? The authors should also cite the data and add a link for readers to access the data. 9. Are there other important tobacco control policies implemented during 2008-2017 in The Gambia? The authors should discuss the policy context in the study period.
--	---

	10. Is cigarette the most commonly used tobacco product in The Gambia? What is the prevalence of novel nicotine and tobacco products (such as e-cigarettes) among youth in The Gambia? 11. Page 12 line 40 “The excise tax typically increases the retail price of cigarettes, thereby reducing the propensity to experiment smoking.” Add citations. 12. Page 13 line 22 “We also found that children are more likely to experiment with cigarette smoking if they are see their teachers smoking in school buildings.” Does The Gambia have smokefree indoor air laws? Might be worthwhile adding some information about smokefree policies (or lack thereof) when discussing this research finding. 13. Page 13 line 43 “The likelihood of smoking increases exponentially.” Do you mean “The likelihood of smoking increases exponentially” as age increases? 14. Page 14 line 15 Misreporting of own smoking behaviors could also be attributed to stigma around smoking. Is there evidence that shows recall bias are more of a concern among older respondents relative to youth? If so, the authors should cite the relevant studies here. References: Chisha Z, Janneh ML, Ross H. Consumption of legal and illegal cigarettes in the Gambia. Tobacco Control 2020;29:s254-s259. Isatou K Jallow, MSc, John Britton, MD, Tessa Langley, PhD. Prevalence and Determinants of Susceptibility to Tobacco Smoking Among Students in The Gambia. Nicotine & Tobacco Research, Volume 21, Issue 8, August 2019, Pages 1113–1121. DOI: 10.1093/ntr/nty128 Cham, B.; Scholes, S.; Groce, N.E.; Mindell, J.S. Prevalence and Predictors of Smoking among Gambian Men: A Cross-Sectional National WHO STEP Survey. Int. J. Environ. Res. Public Health 2019, 16, 4719. DOI: 10.3390/ijerph16234719 Jallow IK, Britton J, Langley T. Prevalence and determinants of tobacco use among young people in The Gambia. BMJ Global Health 2017;2:e000482. Cham, B.; Mdege, N.D.; Bauld, L.; Britton, J.; D’Alessandro, U. Exposure to Second-Hand Smoke in Public Places and Barriers to
--	---

	the Implementation of Smoke-Free Regulations in The Gambia: A Population-Based Survey. Int. J. Environ. Res. Public Health 2021, 18, 6263. DOI: 10.3390/ijerph18126263 Isatou K Jallow, Msc, John Britton, MD, Tessa Langley, PhD. Exploration of Policy Makers' Views on the Implementation of the Framework Convention on Tobacco Control in the Gambia: A Qualitative Study. Nicotine & Tobacco Research, Volume 21, Issue 12, December 2019, Pages 1652–1659. DOI: 10.1093/ntr/ntz003
--	---

REVIEWER	Martin, Ruzima
REVIEW RETURNED	09-May-2022

GENERAL COMMENTS	Overall, this is original, well designed and written manuscript. The introduction is relevant and provides enough information. Adequate information about the previous similar studies have been reviewed and the gap was identified. The appropriate methods, such as survival analysis, were chosen and applied to the data. Overall, the results are accurately interpreted and compared to the previous studies in the area. Overall, this is a high quality manuscript that has implications for the theoretical basis as well as a contribution to the public health policy formulation in Gambia. Specific comments follow:  1. On page 6, author(s) stated that “there is little evidence on the impact of prices on tobacco use experimentation among the young people in the Gambia.” Please cite some of the related works that have been conducted on the topic otherwise rephrase your statement and indicate that there are no similar studies conducted in Gambia. 2. On page 2 on line 19-20. “We also found that children are more likely to experiment with cigarette smoking if they are see their teachers smoking in school buildings.” Please add “ing form” to the verb “to see”. 3. On page 4. Author (s) mentioned that “cigarette prices remain relatively low” and have briefly discussed the background of the country; I would suggest to add a short information (one or two sentences are enough) on the per capital income. This will help to understand how the current price is low among Gambian.
--

VERSION 1 – AUTHOR RESPONSE

Reviewer: 1
Dr. Shaoying Ma, OSUMC
Comments to the Author:

It is with pleasure that I review the paper titled “The effect of price on the decision to experiment with cigarette smoking among Gambian children”. The authors use the method of survival analysis, to study the link between cigarette prices and youth (12-17 yo) experimenting with smoking in The Gambia. Data used in this paper include Global Youth Tobacco Survey (GYTS) data, as well as data from the Gambia Bureau of Statistics. The authors find that 1% increase in cigarette price is associated with 0.7% decrease in the likelihood of experimenting with smoking (defined as taking first puff) among youth. They also document other important factors of youth experimenting with smoking in The Gambia, aside from cigarette prices. Furthermore, this study shows gender differential in the behavior of experimenting with smoking among Gambian youth. This study has the important policy

implication that raising excise tax on cigarettes in The Gambia could be effective in preventing youth from experimenting with smoking.

Although there have been studies on the prevalence and factors of cigarette smoking (or tobacco use in general) among adults and youth in The Gambia, this study contributes to the literature by examining cigarette price as a significant factor of Gambian youth experimenting with smoking, and it has important policy implication on increasing taxation to prevent youth initiation of cigarette use.

Please see below for specific comments. The page numbers are based on those at the lower right corner of the manuscript.

Reviewer's comment:

1. Page 4 line 21

"... this is the second-largest percentage increase after the Eastern Mediterranean region."

Consider rephrasing this sentence to make it more readable, especially because the percentage change was not reported in the sentence.

Authors' response: We have now included the percentage increase to make the sentence flow. The sentence is now structured as follows:

In the WHO African region, the total number of smokers is projected to increase by 27.3% from 66 million in 2015 to 84 million by 2025;³ this is the second-largest percentage increase after the Eastern Mediterranean region.

Reviewer's comment:

2. Page 5 line 3

"The Gambia is a low-income country and is ranked 172 out of 189 countries in the 2020 United Nations Human Development Index."

For the ranking in UN human development index, add citation.

Authors' response: We have now cited the references as follows:

The Gambia is the smallest country on mainland Africa with a population of about 2 million. It is a low-income country and is ranked 172 out of 189 countries in the 2020 United Nations Human Development Index,⁴ with per capita GDP of \$773.⁵

Reviewer's comment:

3. Page 5 line 3

"The country ratified the WHO Framework Convention on Tobacco Control (WHO FCTC) in September 2007. There are several regulations and policies on tobacco control in The Gambia, including the Prohibition of Smoking in Public Places Act 1998, the Tobacco Control Regulations 2019 and the Tobacco Control Act 2016."

Citations need to be added to both sentences.

Authors' response: Both sentences have now been cited as follows:

The country ratified the WHO Framework Convention on Tobacco Control (WHO FCTC) in September 2007.⁶ There are several regulations and policies on tobacco control in The Gambia, including the Prohibition of Smoking in Public Places Act 1998, the Tobacco Control Act 2016 and the Tobacco Control Regulations 2019.^{7 8}

Reviewer's comment:

4. Page 6 line 30

How is junior secondary school defined in The Gambia?

Authors' response: In The Gambia, Junior Secondary are from Grades 7-9 and Senior Secondary Grades 10-12. We have now included that in the manuscript.

Reviewer's comment:

5. Page 6 line 32

I think the authors could add some information here to talk about the representativeness of GYTS sample. Is it a nationally representative sample of junior secondary school students in The Gambia? What is its sampling strategy? Is it a school-based survey?

Authors' response: We have now added the following paragraph:

The GYTS is a nationally representative school-based survey designed by the Centre for Disease Control and Prevention (CDC) as a global standard tool for monitoring tobacco use among youth and to guide the implementation and evaluation of tobacco prevention and control programmes.⁹ The survey considers a cross-section of students in junior secondary schools and does not follow individuals over time, but provides data on their smoking patterns. The sample is drawn using a two-stage cluster-sampling design.^{10 11} Schools are selected with probability proportional to school enrolment size during the first stage, and then classes within participating schools are selected as a systematic equal probability sample with a random start during the second stage.¹¹ All students in the selected classes are eligible to participate in the survey.^{9 11}

Reviewer's comment:

6. Page 6 Table 1 column 2

Is there a reason why 55.7% are girls and 44.3% are boys in the sample of youth with non-missing information on their smoking status?

Authors' response: We find this representation fine as it somewhat corresponds to the gender distribution in the sample before data cleaning (male 44.5%, female 55.4%).

Reviewer's comment:

7. Page 7 line 34

The rationale for choosing survival model as the method of this paper should be provided in the methodology section. Also consider providing that at the end of the introduction section.

Authors' response: We have reformulated part of the first paragraph of the methodology section as follows:

The study employs a duration (or survival) model to estimate the probability of a respondent experimenting with smoking. The survival analysis allows us to analyse the length of time until the occurrence of a well-defined end point of interest,¹² which in this case is smoking experimentation. Experimentation is defined in terms of the first time the respondent smoked (at least part of) a cigarette. It is obtained from the question: "Have you ever tried or experimented with cigarette smoking, even one or two puffs?". The timing of the transition from having never smoked into experimentation depends on the probability of experiencing a transition in period t , conditional on not having experienced a transition until period t ; is also known as the hazard rate or conditional failure rate.² The estimation approach requires longitudinal data, making it necessary to transform the cross-sectional GYTS data into pseudo-longitudinal data.^{1 2 13} The transformation allows for the analysis of time-to-event data. Such data describe the length of time until the occurrence of an event of interest.

Reviewer's comment:

8. Page 8 line 34

More details about the price data from the Gambia Bureau of Statistics need to be provided. Are those data publicly accessible? How was the cigarette price measure calculated by the Bureau of

Statistics? Is it the price of the most popular brand, or something else? Is it the price of 20 sticks in one pack of cigarette?

The authors should also cite the data and add a link for readers to access the data.

Authors' response: The average retail price is for the most popular brands. We have now made it more explicit as follows:

Data obtained from The Gambia Bureau of Statistics¹⁴ show that the average retail price for a packet of 20 sticks (of the most popular brands) ranged between GMD26.15 (US\$0.51) and GMD31.70 (or US\$0.62, using 2019 exchange rates) between 2008 and 2019, which is substantially lower than the sub-Saharan African regional average of US\$1.80, and the global average price of US\$3.82.¹⁵

Yes, it is the price of 20 sticks in one packet of cigarette.

We obtained the data from the The Gambia Bureau of Statistics office. We have now included the reference. However, the dataset is not public, as such there is no web link to access it.

Reviewer's comment:

9. Are there other important tobacco control policies implemented during 2008-2017 in The Gambia? The authors should discuss the policy context in the study period.

Authors' response: We have now included the following paragraph in the introduction section (page 7):

There are several regulations and policies on tobacco control in The Gambia, including the Prohibition of Smoking in Public Places Act 1998, the Tax Policy Reform Act 2013, the Tobacco Control Act 2016 and the Tobacco Control Regulations 2019.^{7,8} These legal instruments seek to protect children from exposure to tobacco products. However, despite the numerous achievements in tobacco control as highlighted above, there are still gaps, especially on the implementation of smoke-free regulation in The Gambia.⁸ There has also been some regression, specifically with the Tax Policy Reform Act, which has been stalled since 2017, following a change in government.

Reviewer's comment:

10. Is cigarette the most commonly used tobacco product in The Gambia? What is the prevalence of novel nicotine and tobacco products (such as e-cigarettes) among youth in The Gambia?

Authors' response: Manufactured cigarettes are the most widely smoked (57.7%), compared with hand-rolled cigarettes (16%), cigars (13.7%) and pipes (12.6%).¹⁶ We do not have data on use of e-cigarettes but we expect it to be very low as it is banned and not popular.

Reviewer's comment:

11. Page 12 line 40

"The excise tax typically increases the retail price of cigarettes, thereby reducing the propensity to experiment smoking."

Add citations.

Authors' response: Two references have now been cited as follows:

The excise tax typically increases the retail price of cigarettes, thereby reducing the propensity to experiment with smoking.^{2,17}

Reviewer's comment:

12. Page 13 line 22

"We also found that children are more likely to experiment with cigarette smoking if they are see their teachers smoking in school buildings."

Does The Gambia have smokefree indoor air laws? Might be worthwhile adding some information about smokefree policies (or lack thereof) when discussing this research finding.

Authors' response: We have restructured the first paragraph of Page 4 as follows:

... There are several regulations and policies on tobacco control in The Gambia, including the Prohibition of Smoking in Public Places Act 1998, the Tobacco Control Act 2016 and the Tobacco Control Regulations 2019.^{7 8} These legal instruments protect children from exposure to tobacco products. Based on these international treaties and local regulations, the country has an obligation to protect children from tobacco exposure. However, despite the numerous achievements in tobacco control as highlighted above, there are still gaps especially on the implementation of smoke-free regulation in The Gambia.⁸

In the Discussion Section, we have expanded the sentence as follows:

We also found that children are more likely to experiment with cigarette smoking if they see their teachers smoking in school buildings. This makes it imperative for the authorities to close gaps on the implementation of smoke-free regulation.

Reviewer's comment:

13. Page 13 line 43

"The likelihood of smoking increases exponentially."

Do you mean "The likelihood of smoking increases exponentially" as age increases?

Authors' response: Yes, we meant exponentially as age increases. We have corrected that.

Reviewer's comment:

14. Page 14 line 15

Is there evidence that shows recall bias are more of a concern among older respondents relative to youth? If so, the authors should cite the relevant studies here.

Authors' response: Yes, recall bias is more prevalent among adult respondents than among the youth. We have now included two references that support that position. The paragraph is as follows:

... For instance, the study relied on individuals' own memory regarding past smoking behaviour. As such, the dependent variable could be subject to measurement error with an unknown bias. However, this challenge is more prevalent among adult respondents.^{1 2} Considering that our sample was composed of children, with relatively short smoking histories, we believe that our findings are robust.

Reviewer: 2

Ruzima Martin

Comments to the Author:

Overall, this is original, well designed and written manuscript. The introduction is relevant and provides enough information. Adequate information about the previous similar studies have been reviewed and the gap was identified. The appropriate methods, such as survival analysis, were chosen and applied to the data. Overall, the results are accurately interpreted and compared to the previous studies in the area. Overall, this is a high quality manuscript that has implications for the theoretical basis as well as a contribution to the public health policy formulation in Gambia. Specific comments follow:

Reviewer's comment:

1. On page 6, author(s) stated that “there is little evidence on the impact of prices on tobacco use experimentation among the children in the Gambia.” Please cite some of the related works that have been conducted on the topic otherwise rephrase your statement and indicate that there are no similar studies conducted in Gambia.

Authors' response: We have rephrased the sentence as follows:

...according to our knowledge, there is no evidence on the impact of prices on tobacco use experimentation among the children in the Gambia.

Reviewer's comment:

2. On page 2 on line 19-20. “We also found that children are more likely to experiment with cigarette smoking if they are see their teachers smoking in school buildings.” Please add “ing form” to the verb “to see”.

Authors' response: The sentence has now been structured as follows:

We also found that children are more likely to experiment with cigarette smoking if they see their teachers smoking in school buildings

Reviewer's comment:

3. On page 4. Author (s) mentioned that “cigarette prices remain relatively low” and have briefly discussed the background of the country; I would suggest to add a short information (one or two sentences are enough) on the per capita income. This will help to understand how the current price is low among Gambian.

Authors' response: We have rephrased the last paragraph on Page 4 as follows:

The Gambia is the smallest country on mainland Africa with a population of about 2 million. The Gambia is a low-income country and is ranked 172 out of 189 countries in the 2020 United Nations Human Development Index,⁴ with per capita GDP of \$773.⁵

References (Please note that the reference numbers are dynamic. So, the reference numbers on this document are different from those on the main manuscript).

1. Asare S, Stoklosa M, Drope J, et al. Effects of Prices on Youth Cigarette Smoking and Tobacco Use Initiation in Ghana and Nigeria. *International journal of environmental research and public health* 2019;16(17):3114.
2. Dauchy E, Ross H. The effect of price and tax policies on the decision to smoke among men in Kenya. *Addiction* 2019;114(7):1249-63.
3. World Health Organization. WHO global report on trends in prevalence of tobacco smoking 2000-2025: World Health Organization 2018.
4. United Nations Development Programme. Human Development Insight 2022 [Available from: <https://hdr.undp.org/data-center/country-insights#/ranks> accessed 27 June 2022.
5. World Bank. GDP per capita (current US\$) - Lower middle income, Gambia, The. 2022 [Available from: <https://data.worldbank.org/indicator/NY.GDP.PCAP.CD?locations=XN-GM> accessed 28 June 2022.
6. United Nations. United Nations Treat Collections. 2022 [Available from: https://treaties.un.org/pages/ViewDetails.aspx?src=TREATY&mtdsg_no=IX-4&chapter=9&clang=en accessed 27 June 2022.
7. World Health Organization. Legislation by country: Gambia 2022 [Available from: <https://www.tobaccocontrolaws.org/legislation/country/gambia/summary#:~:text=The%20law%20prohibits%20the%20sale,worship%2C%20and%20other%20specified%20locations.> accessed 27 June 2022.
8. Cham B, Mdege ND, Bauld L, et al. Exposure to Second-Hand Smoke in Public Places and Barriers to the Implementation of Smoke-Free Regulations in The Gambia: A Population-Based Survey. *International journal of environmental research and public health* 2021;18(12):6263.

9. Centers for Disease Control and Prevention. Global Tobacco Surveillance System Data. 2021 [Available from: <https://nccd.cdc.gov/GTSSDataSurveyResources/Ancillary/DataReports.aspx?CAID=2> [Accessed: 23 January 2021].
10. Cadmus EO, Ayo-Yusuf OA. The effect of smokeless tobacco use and exposure to cigarette promotions on smoking intention among youths in Ghana. *Cogent Medicine* 2018;5(1):1531459.
11. Boachie MK, Immurana M, Tingum EN, et al. Effect of relative income price on smoking initiation among adolescents in Ghana: evidence from pseudo-longitudinal data. *BMJ open* 2022;12(3):e054367.
12. Schober P, Vetter TR. Survival analysis and interpretation of time-to-event data: the tortoise and the hare. *Anesthesia and analgesia* 2018;127(3):792.
13. Vellios N, van Walbeek C. Determinants of regular smoking onset in South Africa using duration analysis. *BMJ open* 2016;6(7)
14. The Gambia Bureau of Statistics. 2020.
15. World Health Organization. WHO report on the global tobacco epidemic 2019: Offer help to quit tobacco use. Geneva: World Health Organization, 2019.
16. Chisha Z, Janneh ML, Ross H. Consumption of legal and illegal cigarettes in the Gambia. *Tobacco control* 2020;29(Suppl 4):s254-s59.
17. National Cancer Institute and World Health Organization. The Economics of Tobacco and Tobacco Control: National Cancer Institute Tobacco Control Monograph 21 NIH Publication No 16-CA-8029A: Bethesda, MD: U.S. Department of Health and Human Services, National Institutes of Health, National Cancer Institute; and Geneva, CH: World Health Organization, 2016.

VERSION 2 – REVIEW

REVIEWER	Ma, Shaoying OSUMC, Center for Tobacco Research
REVIEW RETURNED	27-Jul-2022

GENERAL COMMENTS	The authors made significant changes which much improved their manuscript. Please see below for one minor comment. The page number is based on those at the lower right corner of the manuscript. Page 5 line 47, “Data obtained from The Gambia Bureau of Statistics[30] show that the average retail price for a packet of 20 sticks (of the most popular brands) ranged between GMD26.15 (US\$0.51) and GMD31.70 (or US\$0.62, using 2019 exchange rates) between 2008 and 2019, which is substantially lower than the sub-Saharan African regional average of US\$1.80, and the global average price of US\$3.82.[31]” I appreciate that the authors add important information about the price data. Please also add the specific year for the exchange rate of GMD26.15 to US\$0.51. I assume it's 2008 exchange rate?
--

VERSION 2 – AUTHOR RESPONSE

Reviewer's comment:

Page 5 line 47, "Data obtained from The Gambia Bureau of Statistics[30] show that the average retail price for a packet of 20 sticks (of the most popular brands) ranged between GMD26.15 (US\$0.51) and GMD31.70 (or US\$0.62, using 2019 exchange rates) between 2008 and 2019, which is substantially lower than the sub-Saharan African regional average of US\$1.80, and the global average price of US\$3.82.[31]"

I appreciate that the authors add important information about the price data. Please also add the specific year for the exchange rate of GMD26.15 to US\$0.51. I assume it's 2008 exchange rate?

Authors' response: We have rephrased the sentence as follows:

Using 2019 exchange rates, data obtained from The Gambia Bureau of Statistics[30] show that the average retail price for a packet of 20 sticks (of the most popular brands) ranged between GMD26.15 (US\$0.51) and GMD31.70 between 2008 and 2019, which is substantially lower than the sub-Saharan African regional average of US\$1.80, and the global average price of US\$3.82.[31]